

# Multiple ecosystem services from field margin vegetation for ecological sustainability in agriculture: scientific evidence and knowledge gaps

Prisila A. Mkenda[1,2,3], Patrick A. Ndakidemi[1], Ernest Mbega[1], Philip C. Stevenson[4,5], Sarah E.J. Arnold[4], Geoff M. Gurr[2,6] and Steven R. Belmain[4]

[1] Department of Sustainable Agriculture, Biodiversity and Ecosystems Management, Nelson Mandela African Institution of Science and Technology, Arusha, Tanzania
[2] School of Agricultural and Wine Sciences, Charles Sturt University, Wagga Wagga, NSW, Australia
[3] Department of Biosciences, Sokoine University of Agriculture, Morogoro, Tanzania
[4] Natural Resources Institute, University of Greenwich, Chatham Maritime, Kent, UK
[5] Royal Botanic Gardens, Kew, Richmond, Surrey, UK
[6] Institute of Applied Ecology, Fujian Agriculture and Forestry University, Fuzhou, China

Corresponding author
Steven R. Belmain,
s.r.belmain@greenwich.ac.uk

## ABSTRACT

**Background:** Field margin and non-crop vegetation in agricultural systems are potential ecosystem services providers because they offer semi-natural habitats for both below and above ground animal groups such as soil organisms, small mammals, birds and arthropods that are service supplying units. They are considered as a target area for enhancing farm biodiversity.

**Methodology:** To explore the multiple potential benefits of these semi-natural habitats and to identify research trends and knowledge gaps globally, a review was carried out following the Preferred Reporting Items for Systematic Reviews and Meta-Analyses guidelines. A total of 235 publications from the year 2000 to 2016 in the Scopus and Web of Science databases were reviewed.

**Results:** The literature showed an increasing trend in the number of published articles over time with European studies leading in the proportion of studies conducted, followed by North America, Asia, South America, Africa and Australia. Several functional groups of organisms were studied from field margin and non-crop vegetation around agricultural lands including natural enemies (37%), insect pests (22%), birds (17%), pollinators (16%), soil macro fauna (4%) and small mammals (4%). Ecosystem services derived from the field margin included natural pest regulation, pollination, nutrient cycling and reduced offsite erosion. Some field margin plants were reported to host detrimental crop pests, a major ecosystem dis-service, potentially leading to increased pest infestation in the field.

**Conclusion:** The majority of studies revealed the importance of field margin and non-crop vegetation around arable fields in enhancing ecosystem biodiversity. Promotion of field margin plants that selectively enhance the population of beneficial organisms would support sustainable food security rather than simply boosting plant

diversity. Our analyses also highlight that agro-ecological studies remain largely
overlooked in some regions.

**Subjects** Agricultural Science, Biodiversity, Ecology, Ecosystem Science, Natural Resource
Management
**Keywords** Agro-ecological intensification, Biological control, Predation, Insect–plant interactions,
Sustainable agriculture, Biodiversity

# INTRODUCTION

The world population is currently 7.7 billion (*United Nations, Department of Economics
and Social Affairs, 2019*) and it is projected to grow to 9.5 billion in 2050 (*Lal, 2015*) and
more than 12 billion by the end of the 21st century, with most of the increase expected to
occur in Africa (*Gerland et al., 2014*). Consequently, food demand will escalate (*Valin
et al., 2014*); however, agricultural intensification through monocultured cropping systems
is not a promising strategy for future needs due to adverse environmental effects (*Jonsson
et al., 2012*; *Robinson & Sutherland, 2002*). In addition, conversion of natural and
semi-natural habitats to arable farms with increased chemical inputs are among the threats
to sustainable agriculture (*Meehan et al., 2011*). Agricultural intensification has replaced
much of the native vegetation across the world and it is estimated about 70% of tropical
land is under agriculture and/or pasture modified systems (*McNeely & Scherr, 2003*;
*Ordway, Asner & Lambin, 2017*). Intensive agricultural systems are associated with
negative environmental impacts, including decreased biodiversity of wild plants and
animals. This can lead to increased pest damage as a result of decline in natural pest
control often caused by increased chemical inputs (*Jonsson et al., 2012*) whilst promoting
pest abundance through monoculture cropping systems (*Meehan et al., 2011*). Various
approaches can be taken to mitigate these impacts, including the adoption of intercropping
(*Martin-Guay et al., 2018*). However, the focus on field manipulation might be insufficient
to increase biodiversity of the farmland throughout the year unless it is supplemented
with proper management of the field margins (*Wiggers et al., 2016*).

In most farmland, field margin vegetation may represent the key semi-natural habitat
available to enhance biodiversity. Field margin abundance, location and management
practices can determine the environmental benefits obtained. Field margins can be
managed for provision of multiple ecosystem services such as medicinal products (*Rigat
et al., 2009*), reduced soil erosion and/or nutrient runoff (*Sheppard et al., 2006*), increased
litter decomposition (*Smith et al., 2009*) and reduced air and water pollution from
runoff and pesticide spray drift (*Sheppard et al., 2006*). Other benefits include increased
biodiversity of different plant and animal groups with various environmental benefits.
Field margins at the boundary of sensitive features like watercourses can provide
additional environmental benefits like protection of water sources from soil erosion and
agricultural pollutants compared with field margin that separates two arable farms
(*Hackett & Lawrence, 2014*). In addition, field margins can serve as habitat corridors to
connect other remnant semi-natural habitat fragments such as woodlands (*Marshall &
Moonen, 2002*). In terms of management, field margins can promote more diverse
organisms when there is also reduced pesticide use, tillage and enhanced crop cover compared with a conventionally managed crop (*Vickery, Feber & Fuller, 2009*). Field margins can be designed to provide a particular benefit for a particular group of organisms. Increased numbers of aerial insects, which are the target food for black-tailed godwit chicks, can be supported through management of field margins of intensively managed grass fields (*Wiggers et al., 2016*). Likewise, *Rouabah et al. (2015)* and *Woodcock et al. (2008)* observed positive responses of carabid beetle distribution and diversity as a result of different management levels of the field margins that increased sward architectural complexity through combinations of inorganic fertilizers, grazing and cutting at different heights and time. *Ramsden et al. (2015)* reported on the potential of field margins for food provisioning, overwintering sites and hosts to various predators and parasitoids for enhanced biological control services in agro-ecosystems. Several studies have reported on the importance of field margin management in arable fields for the provision of foraging habitats, nesting sites, food resources and shelter for invertebrates and vertebrates (*Bianchi, Booij & Tscharntke, 2006*; *Gurr, Wratten & Luna, 2003*; *Landis, Wratten & Gurr, 2000*; *Marshall, 2004*). These benefits can be particularly important after disturbances caused by agricultural practices like tillage, pesticide application and harvesting (*Lee, Menalled & Landis, 2001*). Field margin establishment and management is one of the affordable measures by a majority of the farmers due to the associated multiple benefits including biodiversity, conservation and functional values (*Moorman et al., 2013*). Understanding the various benefits of field margin and non-crop vegetation in agriculture and the environment is particularly important for proper management.

Field margins consist of native and/or non-native plants that separate the cropped area from hedgerows or other off crop features. Broadly, field margins are grouped under two major categories: cropped field margins and uncropped field margins (*Vickery, Feber & Fuller, 2009*). Cropped field margins contain sown arable crops that are identified using ecological and conservation principles. Margins can be managed using the existing field operations where the cultivated strip land is left to regenerate naturally or planting strips to provide food resources to insects. Uncropped field margins are set aside margins that are sown (with wild seed mixtures) or left to regenerate naturally without human manipulation. Both cropped and uncropped field margins can be maintained in various ways including cutting to reduce shading and invasion to the field.

Field margins may provide various environmental benefits depending on the establishment and management method employed (*Bowie et al., 2014*; *Fritch et al., 2011*; *hUallacháin et al., 2014*; *Meek et al., 2002*; *Vickery, Carter & Fuller, 2002*; *Walker et al., 2007*). For example, uncropped margin types were found to be more capable of supporting high plant density compared with cropped field margins, due to the effect of competition from the crop (*Walker et al., 2007*). Multiple benefits may be achieved where different margin types are incorporated at the same farm because no single field margin is capable of providing the required food and habitat resources to all plants and animal groups (*Olson & Wäckers, 2007*; *Vickery, Feber & Fuller, 2009*; *Woodcock et al., 2009*). Establishment and management method employed upon the field margin in arable farmland (Fig. 1) may significantly influence the long term conservation values (*Smith et al., 2010*). Therefore, the

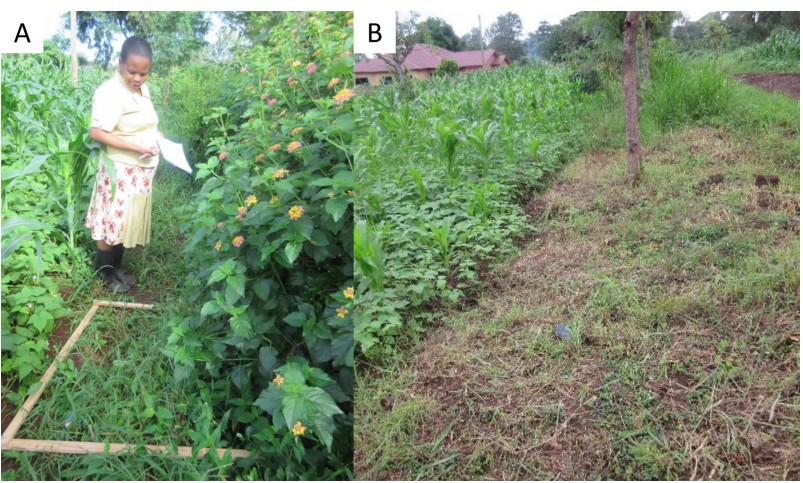

Figure 1 **Field margin management practices, undisturbed (A) and disturbed (B).** Undisturbed field margin vegetation around agricultural lands are useful in provision of nectar and habitat for beneficial arthropods thereby enhancing ecosystem services. Disturbed or cleared field margins are less efficient in enhancing beneficial arthropods. Photo credit: Patrick Ndakidemi.

intention of integrating agronomic and biodiversity objectives may be achieved through field margin establishment and management.

## SURVEY METHODOLOGY

The objective of the study was to analyze the multifunctional role of field margin and non-crop vegetation in agriculture and to identify research trends and knowledge gaps in the world by reviewing published articles. The review was carried out following the Preferred Reporting Items for Systematic Reviews and Meta-Analyses guidelines (*Moher et al., 2009*) and focused on both geographical and temporal distribution of the studies published in the year 2000–2016. The literature was accessed from Scopus scientific database using a series of key words: "field margin*" OR "non crop*" OR "margin plant*" OR "border plant*" OR "margin vegetation*" in the subject area of agricultural, biological and environmental sciences.

A total of 1,153 research articles, 63 review papers and 54 conference papers containing the key words in title, abstract or keywords were found. These items were trimmed to 204 research articles, five review papers and eight conference papers, making a total of 217 publications based on the criterion that the search terms appeared in the title. A further search using the same key words in the title from Web of Science database led to 197 research articles and 10 proceedings papers. These publications were then crosschecked between the two databases to avoid duplications, adding eight research articles and 10 conference papers/proceedings as the only additional materials from the Web of Science database. This brought the total number of publications considered in this review to 235. Detailed analysis of the literature was done to extract information on the spatial data (study location), animal groups studied and ecosystem services and disservices derived

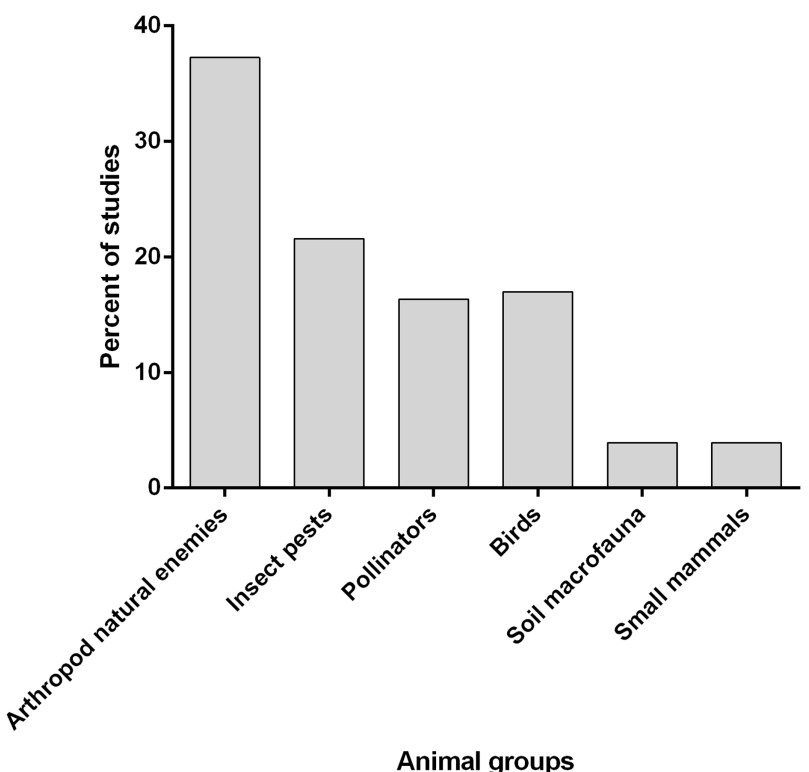

**Figure 2 Animal groups that benefit from the field margin and non-cropvegetation around agricultural lands.**

from the field margin biodiversity. Information on the impact of farming and management practices to the field margin flora and diversity was also analyzed.

## RESULTS AND DISCUSSION

There has been a marked increase in the numbers of publications from 2000 to 2016. These studies were largely conducted in European countries followed by North America then Asia, South America, Africa and Australia. The animal groups studied include arthropod natural enemies, insect pests, pollinators, birds, soil macrofauna and small mammals (Fig. 2).

Other studies assessed the environmental factors (such as landscape structure, hedge stand types and site conditions) that determine the flora composition of field margins (*Guiller et al., 2016*; *Sitzia et al., 2013*; *Sitzia, Dainese & McCollin, 2014*; *Wrzesień & Denisow, 2016*; *Street et al., 2015*). The role of field margins in preventing soil erosion (*Ali & Reineking, 2016*; *Sheppard et al., 2006*) and soil carbon losses (*D'Acunto, Semmartin & Ghersa, 2014*; *Falloon et al., 2004*) were also studied. It was further reported that field margins are ecologically affected by the agronomic and management practices employed within the crop land like pesticide, herbicides and fertilizer application (*Alignier & Baudry, 2015*; *Hahn, Lenhardt & Brühl, 2014*; *Kang et al., 2013*; *Schmitz, Schäfer & Brühl, 2013*; *Schmitz, Hahn & Brühl, 2014*; *Schmitz, Schäfer & Brühl, 2014*). The ecological effects of field margin plants on weed infestation in the field (*De Cauwer et al., 2008*;

*Reberg-Horton et al., 2011*; *Tarmi, Helenius & Hyvönen, 2011*) and organic matter decomposition (*Smith et al., 2009*) were also investigated.

## Multiple benefits of field margin and non-crop vegetation around arable farms

According to *Smith et al. (2008)*, field margins play three major ecological roles including enhancing biodiversity, provision of habitat refuge for rare and endangered species and promoting ecosystem services like natural pest regulation, pollination and nutrient cycling. These three ecological benefits of the field margin may be categorized as biodiversity value, conservation value and functional value respectively. This is apparent from the literature reviewed as most of the studies were related to biodiversity and functional values while only a few focussed on conservation value, particularly habitat and food resource provision to rare and endangered bird species.

### Enhancement of arthropod natural enemies and biological control of insect pests

From the literature review, natural enemies were the most studied in terms of the number of publications compared with other groups. The most studied natural enemies were spiders and ground beetles (Carabidae) since these organisms are regarded as biological indicators in biodiversity and conservation assessments as well as indicators of change in terrestrial ecosystems (*Perner & Malt, 2003*). Other natural enemies studied were ladybirds (Coccinellidae), hover flies (Syrphidae), tachinid flies (Tachinidae), predatory bugs (including Miridae, Reduviidae), parasitoid species of various families (Chalcidoidea, Ichneumonoidea, Chrysidoidea and Proctotrupoidea), Neuroptera and ants (Formicidae) (*Anderson et al., 2013*; *Balzan, Bocci & Moonen, 2016*; *Bowie et al., 2014*). The studies supported hypotheses about the importance of increased diversity of field margin plants and landscape complexity to the populations of different natural enemy groups and pest control (*Atakan, 2010*; *Pluess et al., 2010*, *Rouabah et al., 2015*; *Torretta & Poggio, 2013*; *Werling & Gratton, 2008*). Strips and borders of non-crop vegetation were found to increase the abundance and diversity of spider communities and other natural enemies (*Amaral et al., 2016*; *Ditner et al., 2013*; *Gurr et al., 2016*; *Pluess et al., 2010*). Field margin plants such as trees and shrubs are considered as refuge sites for increased population of predatory insects (*Burgio et al., 2004*). It was found that field margins with several plant species at local and landscape level are effective in managing pests compared with simplified field margins (*Bischoff et al., 2016*). Field margins with sufficient flowering plants act as reservoirs of beneficial insects to recolonize the crop field as observed in hoverflies and tachinids (*Inclán et al., 2016*; *Sutherland, Sullivan & Poppy, 2001*). They are also regarded as hotspots for other beneficial insects including ground beetles as an indicator species (*Eyre et al., 2016*; *Yu, Liu & Axmacher, 2006*). Attractiveness of the flowers and presence of nectar are reported to be the major factors that enhance the parasitoid population in the field margin plants (*Bianchi & Wäckers, 2008*). Whiteflies are an example of one taxon found to be effectively controlled by parasitoids that were enhanced as a result of the floral nectar of non-crop vegetation around bean fields

(*Hernandez, Otero & Manzano, 2013*). Non-crop habitats within arable lands thus significantly influence the abundance and diversity of natural enemies. From the literature reviewed, it was found even a very small area (tens of square meters) of non-crop habitat had a significant effect on the population of ground dwelling spiders (*Knapp & Řezáč, 2015*; *Pluess et al., 2010*; *Jung et al., 2008*) and carabid beetles (*Knapp & Řezáč, 2015*; *Marasas, Sarandón & Cicchino, 2010*; *Werling & Gratton, 2008*). Contradictory findings of a much weaker influence of non-crop vegetation on spider populations were reported by *D'Alberto, Hoffmann & Thomson (2012)*, where other factors like crop characteristics (annual vs perennial) and regional differences appeared to play a larger role. Arthropod populations in field annual crops are highly dependent on the surrounding non-crop vegetation because of the periodic disturbances that occur within the field crop unlike the perennial plants where there is less disturbance. Another study by *Noordijk et al. (2010)* reported on the influence of the field margin age to invertebrate population where predators were found to decrease with increase in the age of the field margin as a result of decrease in plant species and species evenness. Generally, many natural enemies are enhanced by timely availability of three key resources: prey as a food resource, floral resources as additional food and shelter habitats and overwintering sites in case of disturbances (*Ramsden et al., 2015*). Some invertebrates move from the field margin to the field crop during the growing season when there is abundant food resources and later back to the margin when the resources are scarce or due to agronomic disturbances (*Girard et al., 2011*; *Sorribas et al., 2016*). This highlights the importance of margin vegetation as alternative shelter and food resource to beneficial insects around crop land.

Additionally, some field margin plants have pesticidal properties which apart from repelling the insect pests in the field, may also be extracted and used as biopesticides and sprayed to the crops to manage pests as reported by *Mkenda et al. (2015)*. The advantage of natural pesticides from plant origin is that they are less likely to harm non-target organisms and the environment in general, particularly due to their low persistence in soil and on surfaces and lower toxicity (*Amoabeng et al., 2013*; *Mkenda et al., 2015*; *Mkindi et al., 2017*; *Tembo et al., 2018*). Many studies have reported on the ecological and economic benefits of botanical pesticides as compared with synthetic pesticides (*Isman, 2006*; *Kamanula et al., 2010*; *Prakash, Rao & Nandagopal, 2008*; *Stevenson et al., 2012*; *Stevenson, Isman & Belmain, 2017*). Therefore, establishing field margins with pesticidal plants is an added advantage that may be particularly beneficial to resource-poor farmers in smallholder or subsistence systems.

Microbial enemies of insect pests in the field margin were also studied in addition to the natural enemies. The transmission of the entomopathogenic fungus *Pandora neoaphidis* in aphids was significantly higher in fields with margins containing several plant species compared with those with just one plant species (*Baverstock, Clark & Pell, 2008*; *Baverstock et al., 2012*). In addition, entomopathogenic fungi are more abundant in soils of organic farms as compared with conventional farms with no significant difference in their field margins (*Klingen, Eilenberg & Meadow, 2002*). Field margins can act as refuge areas during pesticide application in conventionally managed fields and they should be considered as potential habitats to enhance populations of natural enemies in the field for pest control.

### Enhancement of insect pollinators

Pollinators play an important role in ensuring high yield through the pollination services they provide. The most common pollinators studied across the literature reviewed were honey bees (*Apis* spp.), hoverflies, beetles, moths, butterflies and non-*Apis* bees. The importance of field margin vegetation to pollination was modeled in monoculture cropping systems and the models predicted that pollinator abundance in the margin would increase with the availability of different floral resources (*Rands & Whitney, 2010*). Butterflies were found to benefit from the grassy field margin as their potential corridors in agricultural landscapes with increased pollination service (*Delattre et al., 2010*). This is because field margins can act as corridors for pollinators to increase their pollination services (*Altieri, 1999*).

Generally, pollinators are more attracted by the flowering plants rich in nectar and pollen along the field margins compared with non-flower margin plants (*Barbir et al., 2015*; *Carvell et al., 2007*; *Ricou et al., 2014*; *Bäckman & Tiainen, 2002*), though preferences for certain resources do exist among different species. For example, *Apis* bees and non-*Apis* bees are reported to differ in terms of their preferences to floral resources and foraging distance (*Rands & Whitney, 2011*; *Rollin et al., 2013*). A study by *Kütt et al. (2016)* found linear habitats such as field margins and road verges to be less effective in providing quality flower-based ecosystem services because they were low in species richness as compared with permanent grasslands. According to *Denisow & Wrzesien (2015)*, pollination services benefit from margin flower plants located at a distance of less than 1,000 m, or if the field area is less than 10 ha. Availability of floral resources for nectar provision close to cropped land enhances pollinator abundance, with associated increased pollination service. The type of field margin, whether cropped or uncropped, may also influence the insect population in such habitats because of the differences in plant species composition. For example, uncropped field margins with several naturally regenerated wildflower plant species harbored more bumblebees and honey bees as compared with cropped margins (*Kells, Holland & Goulson, 2001*). This shows the need for more research on the influence of different margin characteristics to pollinators and the value of pollination service to crop yield where such studies are limited.

### Increased survival of bird species

Some bird species which have been already identified as threatened species were observed in the field margin of agricultural lands in Europe (*Wuczyński et al., 2014*), flagging the importance of margin habitats. Several measures have been put in place to conserve the rare and endangered bird species, including non agri-biodiversity programs like Agri-Environment Schemes (AES) (*Carvell et al., 2007*; *Field et al., 2007*; *Marshall, West & Kleijn, 2006*; *Merckx et al., 2009*; *Kleijn et al., 2001*; *Tarmi, Helenius & Hyvönen, 2011*; *Smith et al., 2008*; *Walker et al., 2007*). However, the majority of AES are not performing well on biodiversity conservation and enhancing ecosystem services because many of them have considered the entire field and primarily the crop area, with less attention focused on the field margins (*Wiggers et al., 2016*). There is a need to combine both AES and proper field margin management to conserve bird population and diversity (*Kuiper et al., 2013*; *Wiggers et al., 2016*).

The benefits of field margins to the survival of bird chicks are reported by several studies (*Di Giacomo & De Casenave, 2010*; *Kleijn et al., 2001*; *Kuiper et al., 2013*; *Vickery, Carter & Fuller, 2002*; *Wilson et al., 2010*). This is because a larger percentage of the plant species that are used as nesting sites are present in the field margin as compared with the field center in temperate arable farms. The increased plant diversity is associated with increased invertebrate biomass (*Balzan, Bocci & Moonen, 2016*; *Hiron et al., 2015*; *Torretta & Poggio, 2013*; *Woodcock et al., 2007*) which may be useful food resources for birds (*Douglas, Vickery & Benton, 2009*; *Wiggers et al., 2015*; *Ottens et al., 2014*; *Perkins et al., 2002*; *Woodcock et al., 2009*). It is also reported that most of the field margins that were established and managed to promote beneficial insects are used by bird species as overwintering and refuge habitats (*Plush et al., 2013*). The optimal age and size of the field margin are reported to affect the richness and breeding densities of bird species where species richness and territory density increased up to the age of 4–6 years of the field margin, thereafter it started to decline (*Zollinger et al., 2013*). The type of field margin vegetation and their characteristics is another potential factor that may influence bird species (*Holt et al., 2010*; *Lemmers, Davidson & Butler, 2014*; *Zuria & Gates, 2013*). Comparison of three types of field margin vegetation classified according to the volume of tall vegetation showed that a tree lined margin supported the highest abundance and diversity of bird species, followed by shrubs and lastly by open (herbaceous margin) habitats (*Wuczyński et al., 2011*). Set-asides are the most preferred habitats for foraging of birds during breeding as compared with grassland or cereal crop margins (*Zollinger et al., 2013*). Despite the fact that the level of benefits differ between different types of field margin with different management approaches, presence of a field margin did significantly increase the avian biodiversity in arable farms (*Marshall, West & Kleijn, 2006*).

### Enhanced survival of small mammals

Small mammals studied in the context of field margin and adjacent vegetation include the harvest mouse, *Reithrodontomys megalotis* (*Čanády, 2013*; *Sullivan & Sullivan, 2006*), several mole species (Talpidae) (*Zurawska-Seta & Barczak, 2012*), house mouse, *Mus musculus* (*Sullivan & Sullivan, 2006*; *Moorman et al., 2013*), deer mouse, *Peromyscus maniculatus*, Great Basin pocket mouse, *Perognathus parvus* and various vole species (*Sullivan & Sullivan, 2006*). These mammals took advantage of the established and well managed field margins that aimed to enhance beneficial insect abundance and diversity. Though they usually feed on crops and, thus, must be primarily considered as pest organisms, in a broader context, they may influence the abundance of vertebrate predators, especially the birds that feed on small mammals, serving as a foundation for many trophic interactions (*Korpimäki et al., 2005*; *Meserve et al., 2003*). In addition to the ecological interaction they serve, they also help to reduce weed infestation in the field by feeding on the undesirable weed seeds (*Howe & Brown, 1999*).

Most studies dealt with omnivorous rodents, but the European mole is an obligate carnivore, feeding on earthworms and other invertebrates in the soil. Thus it is not considered as a crop pest (*Lund, 1976*). The damage caused by mole is through burrowing activities which leads to molehills that may affect vegetation composition of the area and

cause occasional damage to silage (*Atkinson, Macdonald & Johnson, 1994*). Consequently, it is considered a pest more in ornamental and amenity contexts than agriculture.

### Promoting soil macrofauna and organic matter decomposition

Above-ground biodiversity was most commonly studied while only 5% of papers, all of which were from Europe, considered the effect of field margin management on soil macrofauna such as earthworms (*Crittenden et al., 2015*; *Nuutinen, Butt & Jauhiainen, 2011*; *Roarty & Schmidt, 2013*). Earthworms are affected by agricultural disturbances such as tillage as it influences soil moisture and, over a long time scale, organic matter (*Kuntz et al., 2013*; *Pelosi et al., 2014*; *Smith et al., 2008*) both of which determine habitat favourability for terrestrial annelids. Several studies (*Crittenden et al., 2015*; *Nuutinen, Butt & Jauhiainen, 2011*) reported an increase in earthworm numbers in the field margin strips with reduced tillage as compared with adjacent arable farms. In general, most of the studies reported that field margin management increased underground soil macrofauna population in comparison with arable lands.

Other groups of soil organisms that were enhanced by field margin management include soil predators, herbivores and detritivores in different taxonomic groups as Haplotaxida, Isopoda, Chilopoda, Diplopoda and Coleoptera (*Smith et al., 2008*; *Anderson et al., 2013*). The age of the field margin was also reported to influence soil detritivore communities, where richness and diversity was positively related with the age of the field margin (*Noordijk et al., 2010*). The biodiversity, conservation and functional values of soil macrofauna was enhanced by field margins that were established and managed with the aim of increasing the arthropod population in arable farmlands (*Smith et al., 2008*). This shows the existence of multiple benefits of field margin plants and the need to maximize such benefits.

Soil biodiversity loss as a result of the expansion, intensification and mechanization of agriculture has been recognized as a major challenge to sustainability (*Pulleman et al., 2012*). The soil ecosystem includes many decomposer taxa that are key to soil formation and structure and play a significant role in nutrient cycling with clear consequences for plant growth and soil carbon storage (*Aislabie & Deslippe, 2013*). In intensive agricultural lands, the densities of soil organisms can be low due to use of agrochemicals and frequent agricultural disturbances, with deleterious effects of decomposition of soil organic matter (*Coleman et al., 2002*). A comparative study of litter decomposition by soil macrofauna revealed increased activity of soil organisms with increased litter decomposition along the field margins with less disturbance compared with the more disturbed areas (*Smith et al., 2009*). Field margins are, therefore, providing a contribution to both below and above ground populations of organisms, but undisturbed field margins have higher values in this respect.

### Reduced soil erosion and nutrient loss

Though soil erosion is a natural process, it can be exacerbated by agricultural intensification that turns it into a major environmental challenge (*Uri, 2000*). While the rate of soil erosion in farming systems is very high it remains lower in well managed

field margins and uncultivated areas (*Pimentel et al., 1995*). According to *Zheng (2006)*, changes in vegetation composition like conversion of natural or semi natural habitats to crop land greatly influence soil erosion processes. Soil erosion leads to decreased soil nutrients which are important in plant growth thus affecting agriculture production (*Lal, 2015*). Apart from on-farm effects, soil erosion can have off-farm effects as well, including sedimentation in other areas and water pollution especially if the source is a cultivated area with agro chemical inputs (*Uri, 2000*; *Van Oost et al., 2007*). Soil erosion is severe in intensively cultivated land with high tillage practices, intensive chemical inputs and monoculture systems (*Jonsson et al., 2012*; *Meehan et al., 2011*; *Robinson & Sutherland, 2002*) due to loosening of the soil particles, rendering the surface susceptible to wind and rainfall erosion (*Pimentel et al., 1995*). The soil erosion in intensively managed agriculture land can be reduced through enhanced soil infiltration (a process in which water on the ground surface enters the soil) which can be achieved through vegetative field margins (*Ali & Reineking, 2016*; *Zheng, 2006*). Other measures that can also be employed to reduce soil erosion include conservation agriculture based on crop rotation (*Sun et al., 2018*), mulching (*Lalljee, 2013*) and cover crops (*Durán Zuazo et al., 2006*; *Lal, 2015*).

Field margins are considered effective in eliminating offsite erosion by trapping the sediments that otherwise could have been loaded in the lowland areas including water bodies (*Duzant et al., 2010*; *Sheppard et al., 2006*; *Uri, 2000*). This is also supported by *Tsiouris et al. (2002)* in which most of the fertilizer applied on wheat crops was filtered at the field margins leading to eutrophication of the margin habitats. They reduce the speed of surface runoff and increase soil infiltration depending on the characteristics of the field margin plants and the slope of the land. Different field margin types with different management levels and inclines are reported to have a potential influence of mitigating soil erosion (*Ali & Reineking, 2016*). In intensively managed landscapes, riparian buffer zones (vegetated areas near water ways) play a similar role of filtering agricultural pollutants that could otherwise enter into water bodies thereby affecting the life of aquatic organisms and other associated ecosystem services.

## Influence of field margin and non-crop vegetation on insect pests and plant viruses

Apart from supporting several beneficial insects and other ecosystem services, field margins have an influence on insect pest populations. They may provide habitat and food resources for both insect pests and their natural enemies in agricultural systems. Therefore, an understanding of their ecological interactions including prey–predator interactions, habitat preferences and mobility, as well as their impact on crop production is important for proper management of the field margins (*Tindo et al., 2009*). Fruit flies such as *Drosophila suzukii* (Diptera: Drosophilidae) are among the most studied insect pests of fruits and have several non-crop plant hosts. Consequently, a better understanding of fruit flies' host ranges among plants of the field margin is essential for effective control strategies (*Arnó et al., 2016*; *Kenis et al., 2016*; *Diepenbrock, Swoboda-Bhattarai & Burrack, 2016*). Unlike fruit flies, spider mites in the *Tetranychus* genus have a narrower host range; nonetheless, their presence in the field crop was similarly found to be associated with the

non-crop host plants around the farmland (*Ohno et al., 2010*). Consequently, concerning crops affected by this pest, thought must be given to whether potential hosts are present among the field margin vegetation.

One such example is the case of scale insects on cassava and the infestation dynamics with respect to non-crop vegetation. The insects (*Stictococcus vayssierei*: Stictococcidae) were recorded from several field margin host plants including both native and exotic plant species of the Congo basin (*Tindo et al., 2009*). Thus, field margin plants could be argued to increase the risk of pest outbreaks on the crop in this case. There is thus a strong need to establish and manage the field margin with plant species that selectively enhance the natural enemies and leave the crop less susceptible to insect pests. However, most of the studies that investigated the effect of well managed field margin vegetation on both beneficial and pest insects reported improved biological control of pest species with few, if any, observations of field margins promoting pest issues (*Atakan, 2010*; *Balzan, Bocci & Moonen, 2016*; *Balzan & Moonen, 2014*; *Eyre et al., 2011*; *Fusser et al., 2016*; *Holland et al., 2008*). For example, aphid densities in broccoli plots surrounded by bare margin were found to be more than four times the aphid densities in plots surrounded by mixed weedy vegetation (*Banks, 2000*). This emphasizes the importance of the presence of diverse vegetation in field margins for biological control of insect pests in the field. The presence of prey in non-crop habitats such as field margins may promote the natural enemy population and hence biological control in the field crop. This is in agreement with the study by *Bianchi & Van Der Werf (2004)* who found the availability of non-pest aphids in the non-crop habitats leads to conservation of ladybirds for enhanced biological control. Thrips, aphids and stink bugs damage was reported to be reduced as a result of increased insect natural enemies in different field margin vegetation (*Eyre et al., 2011*; *Alhmedi et al., 2011*; *Pease & Zalom, 2010*). Other insect pests like moth larvae (*Balzan & Moonen, 2014*) and olive psyllids (*Paredes et al., 2013*) were also found to be effectively managed through enhanced biological control attributed to the non-crop vegetation diversity. It is further reported that more than 90% of cereal aphids were effectively controlled in fields with wide margins by flying predators (*Holland et al., 2008*). Further studies on the effect of wildflower strips that were established at the field margin for enhancing beneficial insect population reported no effect on insect pest conservation (*Hatt et al., 2018*).

The information that some field margin plants may be the most preferred host of some pest species or plant disease vector is useful for selection of the most appropriate species of field margin plants for a given system. There are some cases where field margin vegetation is unable to enhance the biological control process due to some factors as summarized in Table 1.

Field margin plants can also be used as trap crops of insect pests, useful in reducing pest populations from the main crop in the field (*Balzan & Moonen, 2014*). Trap crops are plants grown for the purpose of attracting and concentrating the damaging organisms like insect pests and prevent them from reaching the target crop (*Hokkanen, 1991*; *Shelton & Badenes-Perez, 2006*). These trap crops can either be planted in rows within the main crop or planted as field margin plants. In this case, proper selection of border plants is

**Table 1 Factors accounting for ineffective pest regulation of field margin vegetation.**

| Influencing factors | Explanation | Example of species studied | References |
|---|---|---|---|
| Lack of effective natural enemy in the area | Invasive pest species may arrive in an area without their biological control agents, unless they are introduced in the area where they can be enhanced by the vegetation diversity | Migratory locust, *Locusta migratoria* | *Lomer et al. (2001)* |
| Intraguild predation | Predation of the biological control agents by other natural enemies lead to more pest outbreak regardless of the vegetation diversity in the area | Insectivorous birds and wasps | *Martin et al. (2013)* |
| Natural enemy dispersal ability | Field margin vegetation are good in harboring the natural enemies, but poor dispersal of the natural enemies may lead to ineffective pest control within the crop land | Carabid beetles | *Fischer et al. (2013)* |
| Margins with non-crop hosts | Host plants (susceptible plants) at the field margins may provide habitat to insect pests and act as a source of pests in the field | *Drosophila suzukii* and *Stictococcus vayssierei* | *Arnó et al. (2016)*; *Kenis et al. (2016)* and *Tindo et al. (2009)* |
| Planting of susceptible crop variety | Planting of susceptible crop varieties with little or no crop diversification may lead to high pest infestation regardless of the presence of margin vegetation | Pegion pea (*Cajanus cajan*) genotypes and maize | *Dasbak, Echezona & Asiegbu (2012)* and *Poveda, Gómez & Martinez (2018)* |
| Field margin with substitutional resource | Depends on the degree to which the alternative resource is complementary or substitutional for the prey. This may limit pest control in the field | Adult lacewing and aphids | *Robinson & Sutherland (2002)* |
| Improved margin (sown species-rich margin) | Improved (undisturbed) field margin may provide favorable habitats for survival and reproduction of some pests | Slugs | *Eggenschwiler et al. (2013)* |
| The quality of field margin plants | The quality of plant resource mediates positive or negative effects on pest suppression within the crop land | Big-eyed bug (*Geocoris punctipes*) and pea aphids | *Eubanks & Denno (2000)* |

essential as also reported by *Schröder et al. (2015)* in which border plants were used to attract aphids from the field crop and thus reduced viral infection into the field. A particularly well documented example of the importance of margin plant selection is that of push–pull studies where the insect pests are pushed away from the main crop using a repellent intercrop and on to the trap crop at the margin (*Cook, Khan & Pickett, 2007*).

In addition to the influence of the field margin vegetation on insect pests, assessment of how they may act as reservoirs of plant diseases like alfalfa mosaic virus and cucumber mosaic virus and bean infection incidence was conducted by *Mueller, Groves & Gratton (2012)*. The study reported less influence of the alfalfa mosaic virus from the margin plants to bean crop and no association was observed between cucumber mosaic virus in the non-crop and bean infestation in the field. Insect pests are also known to be vectors of several plant diseases, especially those which are caused by virus and bacteria (*Manandhar & Hooks, 2011*). For example, aphids are the main vector in the spread of viruses that cause plant disease. Movement behavior of alate aphids that is aided by wind increases the spreading of the virus and it is often high near the edge of the field as compared with the field center (*Adlerz, 1974*; *DiFonzo et al., 1996*; *Perring et al., 1992*). Therefore, manipulation of the field margin by planting an alternate crop that acts as a screen around the main crop has been found to be effective in crop protection against

**Table 2 Reduced spread of plant viral diseases using border plants as protector plants.**

| Border plants | Main crop | Disease controlled | References |
|---|---|---|---|
| Sunflower | Pepper | Potato Virus Y (PVY) | *Simmons (1957)* |
| Maize | Potatoes | Potato Virus Y (PVY) | *Schröder et al. (2015)* |
| Sorghum, soybean and wheat | Potatoes | Potato Virus Y (PVY) | *DiFonzo et al. (1996)* |
| Bushclover and sunn hemp | Pumpkin | Watermelon Mosaic Virus (WMV) and *Papaya ringspot virus* (PRSV) | *Murphy et al. (2008)* |
| Barley | Broad bean | Bean Yellow Mosaic Virus | *Jayasena & Randles (1985)* |
| Sorghum, corn and vetch | Peppers | Cucumber Mosaic Virus (CMV) and PVY | *Fereres, (2000)* |
| Sorghum | Pumpkin | Watermelon Mosaic Virus (WMV) and Papaya ringspot Virus type-W | *Damicone et al. (2007)* |

non-persistent viral diseases (*Damicone et al., 2007*). The effectiveness of the border plants in managing the spread of disease depends on several factors including the height of the border plants in relation to the main crop. The spread pattern of the virus and the level of preference between the border plant and the main crop by the disease vector may also affect the rate of spread of the disease (*Fereres, 2000*). The border plant may act as a sink or as a physical barrier to the plant virus. As a sink this may be where the infective vector loses the virus when probing non-crop plant species after landing on border plants, by cleansing the mouth parts and reducing the spread of the virus into the adjacent main crop (*DiFonzo et al., 1996*). As a physical barrier this is where the tall border plants simply reduce the possibility of the aphids landing on the adjacent main crop (*Fajinmi & Odebode, 2010*). Table 2 summarizes some of plant borders that were found to be effective in reducing the spread of plant viral diseases into the main crop.

## Influence of field margin and non-crop vegetation on weed infestation in the field

Field margin and non-crop vegetation can become weeds if they spread into the field crop. Many farmers fear weed infestation from the margin in to the field crop, a belief which is often not supported by evidence (*Mante & Gerowitt, 2009*). *Reberg-Horton et al. (2011)* found no evidence of field margin *Amaranthus retroflexus*, *Cyperus esculentus*, *Urochloa platyphylla*, *Ipomoea* sp., *Digitaria sanguinalis*, *Mollugo verticilliata*, *Lamium amplexicaule*, *Sida spinosa* or *Senna abtusifolia* spreading to adjacent maize (*Zea mays* L.) or peanut (*Arachis hypogaea* L.) fields in USA. The type of field margin, plant composition (including their dispersal traits) and distance to the field crop are important factors to consider on whether field margin plants will have an influence on weed infestation in the field (*Blumenthal & Jordan, 2001*). However, different weed species may respond differently to these factors, therefore necessitating the need for an understanding of the specific weed functional traits for effective management (*Reberg-Horton et al., 2011*). For example, the seeds of anemochorous species which are adapted to wind dispersal may disperse only over a short distance (*Feldman & Lewis, 1990*) though spreading of field margin plant seeds that are adapted to wind dispersal is thought to be high and over long distance compared with plant species with no specialized dispersal structure.

Nevertheless, the presence of weeds within the crop is regarded as one of the ways to enhance biodiversity in agro ecosystems (*Clough, Kruess & Tscharntke, 2007*). However,

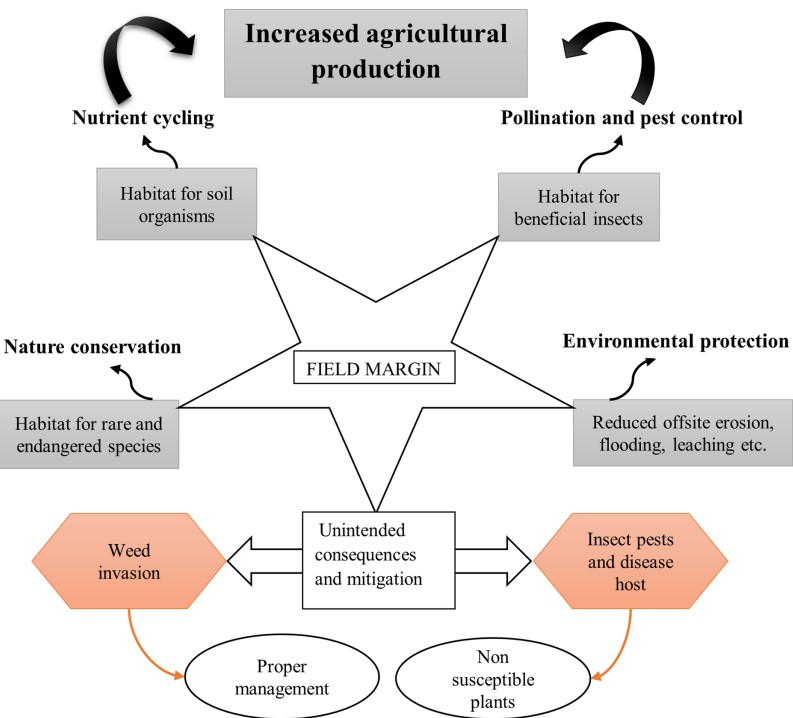

**Figure 3  Potential benefits and dis-benefits derived from field margin vegetation.**

challenges stemming from the competition with crops as well as difficulties during harvesting, especially if mechanized, may arise. From the literature reviewed, a major observation from several studies was that weed dissemination into the field largely depends on the type of margin and the way it is maintained throughout the year. *De Cauwer et al. (2008)* reported on the importance of sown field margin, which are managed through removal of the cuttings in suppression of weed spreading into the field. Similar findings on the importance of sown field margins with proper management for weed control are reported (*West, Marshall & Arnold, 1997*; *Bokenstrand, Lagerlöf & Torstensson, 2004*; *Boutin et al., 2001*). This being the case, it can be concluded that field margin plants are not necessarily the source of weed infestation into the field, and that for weed control, the establishment and management practices on the fields matter most. Major benefits of field margin vegetation as well as possible unintended consequences and mitigation measures are summarized in Fig. 3.

## Agronomic and management factors influencing field margin plant composition

The various management techniques of the field margin and farming operations in the adjacent field have an impact on both field margin flora and fauna composition. Field margin establishment by fencing, application of sown flower mixtures or natural regeneration after the soil is tilled with rotating blades or rotavator (*Fritch et al., 2011*; *hUallacháin et al., 2014*) and their structural connectivity (*Fridley, Senft & Peet, 2009*;

*Kang et al., 2013*) determine their vegetation structure and plant diversity. Field margins established through sowing seed mixtures led to the highest diversity of flora and fauna, especially in highly intensified land (*Fritch et al., 2011*). Subsequent management such as cutting (*De Cauwer et al., 2008*), grazing or mowing (*Coulson et al., 2001*; *Fritch et al., 2011*), coppicing, trimming and pollarding (*Deckers, Hermy & Muys, 2004*) and other techniques including agrochemical input applications (*Schmitz, Hahn & Brühl, 2014*) have been found to influence the floral species composition as a result of disturbance or changes to the soil nutrient content. Field margins may also be affected by weed invasion, if it alters their vegetation structure and composition depending on establishment and management measures employed (*Bokenstrand, Lagerlöf & Torstensson, 2004*; *De Cauwer et al., 2008*; *Reberg-Horton et al., 2011*; *West, Marshall & Arnold, 1997*). Other factors influencing the vegetation structure and composition at the field margin include the ecological and biogeographical context of the area, as well as their historical seedbanks. Field margins have more seedbanks and hence are more species rich compared with the field center (*Jose-Maria & Sans, 2011*).

Likewise, farming activities adjacent to the field margins such as the application of herbicides (*Boutin, Elmegaard & Kjær, 2004*; *Riemens et al., 2009*), pesticides and fertilizers (*Schmitz, Schäfer & Brühl, 2013*; *Schmitz, Schäfer & Brühl, 2014*) can be considered potential disturbances and may adversely affect the margin flora structure and composition. The effect of fertilizers and herbicides significantly affected the occurrence and frequency of several light feeder plant species that require less nitrogen and other nutrients leading to low diversity while few heavy feeders (plant species with high demand of nitrogen and other nutrients) were favored by the applied fertilizer (*Schmitz, Hahn & Brühl, 2014*). Though agrochemical inputs are typically applied in the crop, their effect can be observed in the field margin as a result of direct overspray or spray drift due to their proximity to the field (*Firbank et al., 2008*). The effects of pesticide drift or overspray are more pronounced in narrow field margins, particularly those less than 3 m wide (*Hahn, Lenhardt & Brühl, 2014*). Therefore, field margin plant composition is greatly influenced by the agronomic and management practices which consequently determines faunal composition and hence ecosystem service/disservice.

## RECOMMENDATIONS

Understanding the current status of the biological diversity of field margins and its integration in agriculture, as well as the influence of human agricultural activities on the various organisms within ecosystems is necessary. Only limited information relating to these processes for most tropical areas are available and in some areas the information has been limited to a few sites with relatively similar ecology and management practices (*Gardner et al., 2010*). Africa particularly is well known in terms of its biodiversity (*Duruigbo et al., 2013*), though very little research on the importance of biodiversity in agriculture has been carried out in this region. Despite all the reported benefits of field margin vegetation established mostly in American (*Amaral et al., 2016*; *D'Acunto, Semmartin & Ghersa, 2014*; *Zuria & Gates, 2013*) and European countries (*Guiller et al., 2016*; *Balzan, Bocci & Moonen, 2016*; *Inclán et al., 2016*; *Knapp & Řezáč, 2015*;

*Rouabah et al., 2015*), its adoption in other continents is still low (*Ndemah, Schulthess & Nolte, 2006*). In view of this, we recommend the following actions.

First, there is a need for increased research effort on effective techniques for enhancing on-farm biodiversity in order to promote ecosystem services for sustainability in agriculture across regions of the world where such research is still limited. From the literature reviewed, it was observed that field borders that were managed to promote the abundance and diversity of above ground beneficial insects were found to support other organisms like birds, soil macrofauna and small mammals as an additional benefit. Other reported benefits include regulation of water and nutrient content within the soil (*Ndemah, Schulthess & Nolte, 2006*), maintaining soil and water quality by preventing erosion and runoff (*Ali & Reineking, 2016*; *Sheppard et al., 2006*) and increased organic matter decomposition by soil organisms (*Smith et al., 2009*). The multiple benefits arising from field margins justify the need for more research and promotion of these habitats as part of sustainable agricultural intensification.

Second, raising awareness among the farmers on the ecological and economic effects associated with the misuse of synthetic pesticides. Many farming communities in developing countries are not aware of the hazards associated with the misuse of synthetic pesticides (*Ngowi et al., 2007*; *Kariathi, Kassim & Kimanya, 2016*). Consequently, they are unknowingly killing the natural enemies of insect pests and disrupting the natural pest regulation service with increased pest resistance to most pesticides. The effects of the pesticides applied on crops extends to the field margin plants due to the proximity of the field margins and the crop land and hence affecting the multiple services derived from the field margin (*Firbank et al., 2008*). It is therefore recommended that agrochemical inputs should be selectively applied or restricted completely in order to increase the diversity of both flora and fauna in agricultural landscapes.

Third, purposive efforts towards adoption of field margin establishment and management among the farmers should be employed. One of the obstacles existing among the farmers in the adoption of new technology is the fear that it might interfere with their normal farming practices, as well as the establishment cost of the technology (*Wilson & Hart, 2000*). However, extensive field margins are among the conservation measures that once established require less effort in maintaining for multiple benefits. Two barriers in some regions may be insufficient knowledge on the ecological benefits of field margins and poor knowledge related to the design of appropriate field margins (*Junge et al., 2009*; *Mante & Gerowitt, 2009*; *Morris, Mills & Crawford, 2000*). These knowledge gaps have led to some difficulties in the acceptance of the intervention among farmers. Social learning and economic incentives such as reduced production cost, more yield, market value or value-added environmental outcome are some of the factors that guarantee wide adoption of an innovation.

Fourth, fulfilling the potential of ecological benefits of semi natural habitats around farm land for improved agriculture and environment requires involvement of various stakeholders (who may vary depending on country) such as farmers, local authorities, researchers, policy makers, NGOs, charities and land or estate owners in the discovery of

the scientific knowledge for easy adoption. Understanding of their personal, social and economic dynamics in the context of innovation adoption is essential.

## CONCLUSIONS

From the literature reviewed, the majority of studies demonstrate that field margin and non-crop vegetation around agricultural lands can provide various benefits including pest control, crop pollination, reduced offsite erosion, organic matter decomposition and nutrient cycling as well as enhancement of rare and endangered species, both above and below ground organisms. Several functional groups of beneficial organisms were reported to benefit from field margin and non-crop vegetation; the most commonly studied were natural enemies, birds, pollinators, soil macrofauna and small mammals. However, some of the field margin plants were reported to host detrimental pests, a major ecosystem dis-service, leading to increased pest infestation in the field. We also identified other factors that are associated with ineffective pest control of field margin vegetation such as lack of natural enemies in the area, intraguild predation, poor dispersal of the natural enemies to the field crop and the overall quality of the field margin vegetation. Therefore, the promotion of field margin plants that selectively enhance the population of beneficial organisms, together with integration of other techniques like use of non-susceptible crops and crop diversification through intercrop would be desirable for sustainability in agriculture.

Though many studies on the role of field margin and non-crop vegetation have been conducted, geographic distribution of the studies is highly skewed. The studies were largely conducted in some countries, especially in Western Europe, but are very limited in number and scope in many tropical countries. The limited research taking place on these semi natural habitats in the tropics may be due to the lack of research funds and poor knowledge on the ecological benefits of these habitats in the agriculture sector in low-income and smallholder farming systems. This calls for the need to raise awareness on the economic and ecological benefits of the semi natural habitats around agricultural fields for sustainable agriculture in areas where farm biodiversity has been given less attention.

## ACKNOWLEDGEMENTS

We thank Charles Sturt University librarians for their support in this study by providing access to the Scopus and Web of Science databases which were used to access the literature reviewed.

### Funding

Steven R. Belmain has received research grants from the McKnight Foundation (#13-335 and #17-070). Geoff M. Gurr has received research funding from the McKnight Foundation (#15-111) and Philip C. Stevenson has received research funding from the BBSRC-Global Challenges Research Fund (BB/R020361/1) and Darwin Initiative

(#22-012). The funders had no role in study design, data collection and analysis, decision to publish, or preparation of the manuscript.

## Grant Disclosures

The following grant information was disclosed by the authors:

Steven R. Belmain has received research grants from the McKnight Foundation: #13-335 and #17-070.

Geoff M. Gurr has received research funding from the McKnight Foundation: #15-111.

Philip C. Stevenson has received research funding from the BBSRC-Global Challenges Research Fund (BB/R020361/1) and Darwin Initiative: #22-012.

## Competing Interests

The authors declare that they have no competing interests.

## Author Contributions

- Prisila A. Mkenda conceived the review and assessed literature, analyzed the data, prepared figures and/or tables, authored or reviewed drafts of the paper, approved the final draft.
- Patrick A. Ndakidemi conceived and designed the experiments, authored or reviewed drafts of the paper, approved the final draft.
- Ernest Mbega conceived and designed the experiments, authored or reviewed drafts of the paper, approved the final draft.
- Philip C. Stevenson conceived and designed the experiments, authored or reviewed drafts of the paper, approved the final draft.
- Sarah E.J. Arnold conceived and designed the experiments, authored or reviewed drafts of the paper, approved the final draft.
- Geoff M. Gurr conceived and designed the experiments, authored or reviewed drafts of the paper, approved the final draft.
- Steven R. Belmain conceived and designed the experiments, prepared figures and/or tables, authored or reviewed drafts of the paper, approved the final draft.

## Data Availability

This is a review article containing no raw data.

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
