# Peer review of "Multiple ecosystem services from field margin vegetation for ecological sustainability in agriculture: scientific evidence and knowledge gaps"

_PeerJ, doi:10.7717/peerj.8091_

## Round 0.1 · original submission · Major Revisions

The referees suggest that the submission may be publishable, but only after some major revisions have been made to your manuscript. Therefore, I invite you to respond to their comments and revise your manuscript.

·

Basic reporting

The main problem throughout the paper is of repetition of information, lack of logical flow especially in the Results and Discussion Section, as well as long (sometimes ambiguous) sentences that make the reading difficult. Also, the writers kept switching between past and present tenses which makes the paper hard to read. Moreover, some facts appear to be contradictory because of poor writing.

Experimental design

The methodology used is adequate and can be replicated but there is a problem with the organization of facts. Although related facts are organized into paragraphs, they are sometimes unnecessarily long with a poor logical flow.

Validity of the findings

Although the paper presents an objective (given in the first few lines of the Methodology Section) with a worldwide view, the conclusions (and scarcely few future directions) are mainly directed toward Africa. Thus, the scope of the paper is broad at the beginning but narrows down to Africa, particularly Tanzania. I think there is not sufficient justification that the recommendations are applicable to "other tropical countries" as presented in the paper.

Additional comments

Though this review paper contains a lot of valuable information and has a broader view (in respect of groups of organisms discussed) compared with previous ones like Vickery et al. (2009), it lacks the elements that will sustain the reader's interest in reading the paper. Because review papers are generally long, readers become bored if they have to read sentences over and over the get the meaning. The paper has many interesting facts but I had to read most sentences two or three times to get the meaning and how they relate to other preceding sentences.

Reviewer 2 ·

Basic reporting

There are some comments in the text in order to improve clarity.

Experimental design

The fact the Africa has little publication/work on the issue should be pointed out in a better way, there are other regions with same gap.

Validity of the findings

All report all well referenced.

Additional comments

Clarify what are the meening of WEEDS and how they differ from NON-CROPs.

Table 1 is not necessary, it can be convert it in text. Figure 2 is already in the text, thus it is not necessary.

Figura 4 should be improved.

Annotated reviews are not available for download in order to protect the identity of reviewers who chose to remain anonymous.

---

## Round 0.2 · Minor Revisions

The referee suggests that the submission may be publishable, but only after minor revisions have been made to your manuscript (see attached PDF). Therefore, I invite you to respond to his comments and revise your manuscript.

·

Basic reporting

Generally good; however, a few sentences have to be moved to different sections to improve the logical flow

Experimental design

No comment

Validity of the findings

No comment

Additional comments

See attached comments.

---

## Round 0.3 · accepted · Accept

I am pleased to inform you that your paper has been accepted for publication.